# Secondary Metabolites Produced by Trees and Fungi: Achievements So Far and Challenges Remaining

Katarzyna Nawrot-Chorabik [1],*, Małgorzata Sułkowska [2] and Natalia Gumulak [1]

1   Department of Forest Ecosystems Protection, Faculty of Forestry, University of Agriculture in Kraków, 29 Listopada Ave. 46, 31-425 Kraków, Poland
2   Forest Research Institute, Braci Leśnej 3, Sękocin Stary, 05-090 Raszyn, Poland
*   Correspondence: k.nawrot-chorabik@urk.edu.pl

**Abstract:** Secondary metabolites are ubiquitous substances occurring naturally in trees and microorganisms. They are produced in various metabolic pathways which determine their structure and biochemical proprieties. However, the biological functions of many secondary metabolites remain undetermined. Usually, the amounts of secondary metabolites produced by trees under natural conditions are limited, which makes their mass production difficult and not cost-effective. Metabolites occurring naturally in plants, including gymnosperm and angiosperm trees, as well as in fungi, are important biologically active substances used by many industries and in modern medicine. The huge variability and potential of biological activity present in secondary metabolites make it possible to replace most of them with compounds of completely natural origin. The current breakdown of metabolites, together with the most important examples of compounds and their uses, are presented in this overview. The possibility of increasing the number of secondary metabolites in a specific environment through interaction with the most known biotic factors is discussed. The use of in vitro culture for the production of secondary metabolites and their extraction, as well as the possibility of subsequent analysis, are described. The current literature on the metabolites produced by individual species is presented.

**Keywords:** secondary metabolites; tree species; fungi species; in vitro cultures; chromatography

## 1. Introduction

Physiological processes conducted by living organisms require both uninterrupted access to energy and suitable sources of macro- and microelements, especially carbon, oxygen, nitrogen, and phosphorus. These are utilized in a variety of chemical reactions arranged in enzymatically controlled metabolic pathways whose entirety is described as cell metabolism (Figure 1) [1].

Basic substances necessary for the growth and reproduction of organisms produced in these reactions are collectively referred to as primary or housekeeping metabolites [2]. However, in some organisms, such as trees, a group of compounds is produced that is not directly necessary for basic cell functions. These are called secondary metabolites. Primary metabolites are necessary for basic metabolic pathways to function; they include structural molecules, compounds involved in energy handling and storage, and molecules necessary for information storage and signaling. Carbohydrates comprise the backbone of nucleic acids and are the main energy source; they also serve as energy storage and structural materials. Triglycerides are the main long-term energy storage material [3] in animals, but also in some plants, especially in seeds of oil plants and in some fruits. They are synthetized in plastids: chloroplasts in the aboveground parts of plants, and proplastids in roots and seeds. The function of proteins is very diverse, as they can include catalytic, structural, signaling, and transfer proteins. The main function of nucleic acids is the storage of genetic information and its expression via synthesis of proteins.

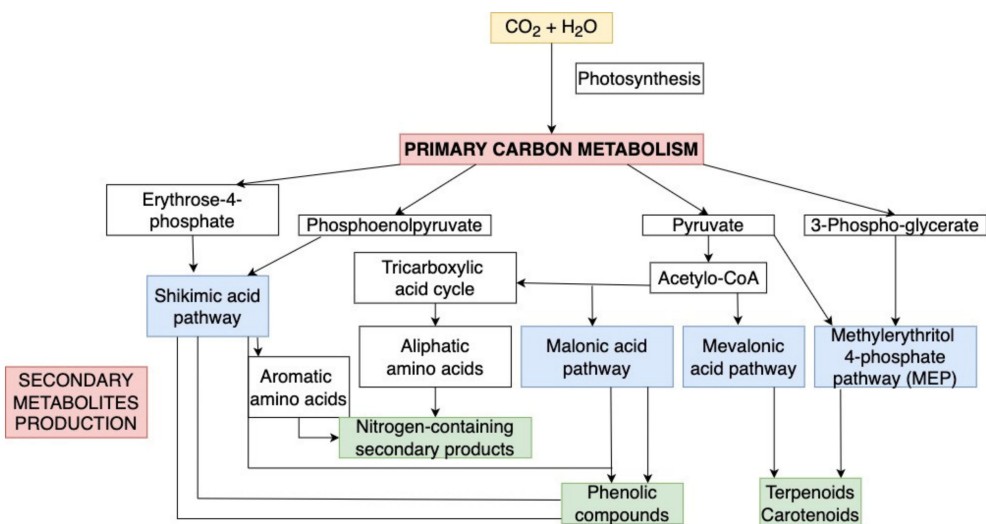

**Figure 1.** Main pathways for the synthesis of secondary metabolites in trees [1].

Unlike the primary metabolites listed above, secondary metabolites are not essential for survival and reproduction. They tend not to occur in all cells of an organism, and are highly variable and species-specific [4]. Unlike animals, cormophyte plants are known for their ability to produce numerous species-specific secondary metabolites. The number of such compounds identified to date reached tens of thousands, but in fact is expected to be much higher. However, the classification of some compounds as either primary or secondary metabolites is debatable, due to the ambiguity of the criteria used to distinguish them. Thus, deciding whether a given compound is a secondary metabolite depends on the recognition of the function it is involved in as essential for basic life processes or not. Moreover, the exact function of most metabolites has not been sufficiently understood yet. This is why, for instance, lignin in some tree tissues is considered a secondary metabolite, but at the same time, it is essential for xylem development, which makes it a primary metabolite. Another example of this difficulty in categorization concerns chlorophyll, which is not necessary to have in order to conduct basic metabolic processes [4].

Particular plant species, as well as plant taxa of higher rank, have their specific profiles of secondary metabolites. In fact, the ability to produce particular metabolites is frequently used as one of the delimitation criteria in tree taxonomy. In early studies on secondary metabolites in trees, they were considered mostly accidental byproducts of primary metabolic pathways [5]. However, an increasing understanding of ecological biochemistry revealed the great advantages these compounds offer their species in competition with other organisms. Still, a sizable number of metabolites has been described whose significance and function remain unknown. Examples of such compounds include polyphenols produced by rubber trees, and some narcotic substances [4]. So far, extensive studies on secondary metabolites have brought forth numerous innovations in the fields of environmental protection [6], pharmacology and pharmaceutics [7], cosmetology [8], food chemistry [9], and agriculture [10]. Interest in studies on secondary metabolites grew especially in the late 19th century after the discovery of the useful properties of some compounds. The role of secondary metabolites includes protection against herbivores or pests, or against pathogens (e.g., arbutin in *Pyrus communis*). They are involved in crosspollination by attracting insect pollinators, or in modifying the color and scent of flowers, leaves, and fruits [11]. They also play a role in plant and tree growth, cellular replenishment, and allocation of resources in plants and trees or their adaptation to local environmental conditions [12].

Another group of secondary metabolite producers is fungi. Their metabolites are characteristically small; the molecular weight of virtually all these compounds does not exceed 1500 Daltons [Da] [13]. They are built mostly of carbon, hydrogen, oxygen, nitrogen,

sulfur, and potassium atoms, but may also include Cl, B, and F. Structurally, they include mostly hydroxyl, carboxyl, and carbonyl groups, but also other structural elements [14].

The aim of this paper is to present the current state of knowledge on secondary metabolites produced by trees and fungi in a condensed form. The methods of increasing the metabolite base in a specific microclimate, as well as the methods that were and are used for their isolation and analysis, are also discussed. In addition, an attempt is made on this basis to demonstrate the new challenges posed by biotechnology and biochemistry to the study of secondary metabolites. No research has been done in the literature on the potential mutual influences of bacteria and fungi, through which the host tree would be able to increase its resistance to biotic and abiotic stresses. Research on the interaction of three organisms from different kingdoms to induce the resistance of one could also improve drug production.

## 2. Main Groups of Secondary Metabolites and Their Characteristics

Secondary metabolites are classified into three main groups (Figure 2) [1]: terpenoids, phenolic compounds, and non-protein nitrogen compounds such as alkaloids [4].

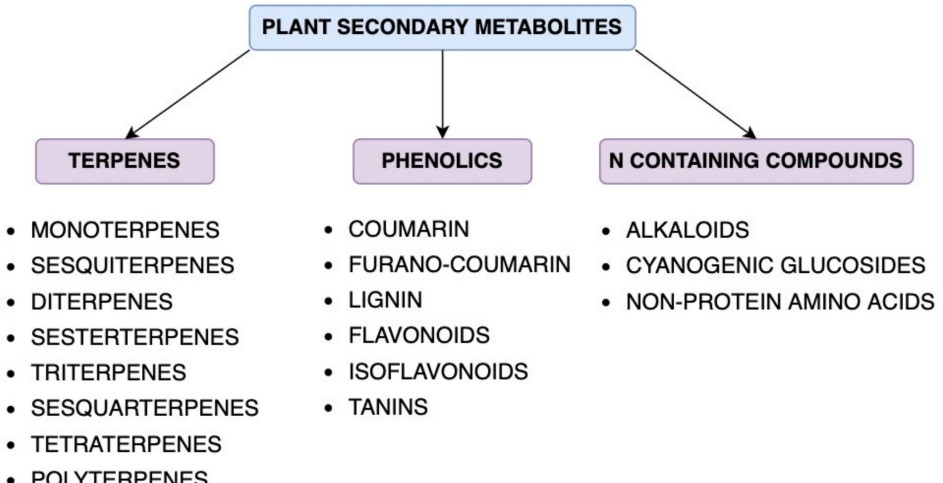

**Figure 2.** The three major types of secondary metabolites [1].

Biosynthesis of terpenoids (also called isoprenoids) requires the activity of either of two metabolic pathways: the mevalonic acid (MVA) pathway [15] or the 1-deoxy-D-xylulose-5-phosphate (DOXP) pathway [16,17]. They are synthetized from five-carbon isopentenyl pyrophosphate (IPP) precursor units or their functional isomer dimethylallyl pyrophosphate (DMAPP). Condensation of IPP and DMAPP in reaction with isoprenyl diphosphate (IDP) results in acyclic, nonchiral isoprenyl diphosphate/pyrophosphate (ID, $C_{5n}$), which is an intermediate precursor for terpenoids [18]. The resulting diversity of terpenoids made up of DMAPP and ID units in various combinations depends on the specificity of various terpene synthases (TPS), and is responsible for their various properties [19].

In plants and trees, one of the more important groups of terpenoids are quinones, such as plastoquinone, occurring in plastids and ubiquinone found in mitochondria. Both of these compounds are actively involved in electron and proton transfer during photosynthesis or cellular respiration. Some photosynthetic pigments, such as carotene and the phytol tails of chlorophyll molecules, also belong to terpenoids. Polyprenol phosphates are responsible for the transfer of polysaccharide chains in plants and trees. Another very important group of terpenoids are sterols, which are main components of cellular membranes, as well as the tree growth regulators gibberellins and abscisic acid. The more subtle role of some terpenoids involves flower pigments, pollinator insect attractors, and participation in seed dispersal. Finally, some terpenoids provide protection against pathogens or herbivores.

However, despite the great prevalence of terpenoids in terrestrial plants and trees, they also occur in algae, fungi, and bacteria [4].

Nomenclature of terpenoids depends on the number of isoprene units in their structures. Monoterpenes, sometimes referred to as monomers, comprise two five-carbon units, while sesquiterpenes contain three units (15 carbon atoms). Accordingly, there are diterpenes, triterpenes, and tetraterpenes; terpenoids comprised of more than 100 carbon atoms are called polyterpenes. An example of polyterpenes are various long-chained isoprene polymers occurring naturally in the sap of rubber trees whose number of units can reach several thousand.

Many terpenoids have unique, valuable properties and are used in various industries. These applications include food additives used to improve fragrance and flavor, or base material for the production of some antibiotics. Moreover, a number of chemical agents used in plant protection were developed based on the structure of terpenoids occurring naturally in some trees. Some mono- and sesquiterpenes found in turpentine are insect repellents, including α-pinene, β-pinene, and camphene, as well as myrcene and limonene occurring in natural resins. Some sesquiterpenes such as glaucolid A or costunolide repel not only insects, but also larger herbivores. Often, such compounds are also categorized as phytoalexins. Phytoalexins are secondary metabolites produced in response to fungal or bacterial infections (directly in defense against the presence of those pathogens or the presence of their metabolites, especially toxins), or even to abiotic stress. These factors are referred to as elicitors. Once triggered, the production of phytoalexins is very rapid due to their low initial concentrations. Numerous phytoalexins are used as insecticides, mostly due to their low environmental impact and low toxicity to mammals. Particular groups of terpenoids serve as intermediate products in the biosynthesis of physiologically important compounds. Sesquiterpenes are precursors of plastoquinone, ubiquinone, abscisic acid, and a variety of phytoalexins [4]. Another set of phytoalexins, gibberellins and phytol, are synthetized using diterpenes. Other diterpenes have deterrent or toxic activity on their own. Some diterpenes are also suspected to have cancer-combating properties, which has recently been the reason for renewed interest in this group of compounds [20]. Triterpenes also include compounds with strong deterrent effects against herbivores, such as limonoids, e.g., azadirachtin. However, the most prominent class of triterpenes is steroids, which are biosynthesized by cyclization of triterpenoid squalene. They include plant sterols, which are analogues of the animal cholesterol in cellular membranes in plants. Some microorganisms are able to use cholesterol, as well as other sterols, as precursors to produce steroid hormones [4].

An important class of terpene-derived secondary metabolites are saponins that are either steroid or tetraterpene glycosides. Most of them are believed to be toxic; due to their amphiphilic nature, they act as detergents disrupting cellular membranes. Some of them also show anti-inflammatory, antiviral, and hemolytic activity [4].

Tetraterpenes include carotenoids and their precursor phytoene. The most prominent among them are red-orange carotenes and orange-yellow xenophiles, which differ only in their numbers of oxygen atoms. Both of these compounds play very important roles in photosynthesis. In addition, some tetraterpenes have potentially beneficial effects on human health, including anticancer activity, prevention of arteriosclerosis, and eyesight improvement [4].

Another very important class of secondary metabolites are phenolic compounds, many of which have not yet been characterized functionally. The common feature of their structure is the occurrence of an aromatic ring and hydroxyl group modified with a various number of substituents (usually carboxyl or methoxy groups). Such a structure is responsible for the polarity of these compounds, which increases their solubility in water. Most naturally occurring phenolic compounds derive from phenylalanine or tyrosine, which are produced via the shikimate pathway [4,21]. The rest are produced in reactions of the malonate pathway. Naturally occurring phenolic compounds can be classified as phenol and its derivatives such as tannins, flavonoids, and isoflavonoids. Bartnik et al.

(2020), who analyzed the concentrations of phenolic compounds in trunks of Norway spruce (*Picea abies*), found a significant difference in their total content between sapwood and heartwood that depended on the level of wood decay. Such a difference gives an opportunity to develop a diagnostic method enabling objective assessment of the quality of spruce wood based on the measurements of the concentration of phenolic compounds [22].

Cinnamic acid, caffeic acid, salicylic acid, and vanillin are examples of simple phenolic compounds found in plants. Cinnamic acid is created from phenylalanine by removing the amino group by phenylalanine ammonia-lyases, and is one of the phenolic acids [23,24]. As they originate from phenylalanine aromatic rings and three-carbon chains, cinnamic acid, as well as its numerous derivatives, are referred to as phenylpropanoids. The role of phenylpropanoids usually involves protection, and many of them can be characterized as phytoalexins as they are synthetized in response to pathogen infection. Some of them are also allelopathic agents [25].

Tannins also derive from phenolic compounds. They can be classified into two groups: hydrolysable tannins produced by polymerization of gallic acid or other gallic-like acids, and carbohydrates, usually glucose, and condensed tannins composed of multiple flavonoid units. The main function of tannins is deterring herbivores, but some of them are toxic due to their ability to bind and denature proteins [26]. This very characteristic is commonly used in the process of tanning leather. Tanning is performed using cleaned and degreased (reduced only to collagen tissue) animal skin or hide in order to maximize tannin binding and collagen denaturation. This process increases leather durability and protects it against decomposition by microorganisms [27]. However, the use of plant tannins is currently limited, as they are largely replaced by chrome tanning, producing better results [28]. Naturally occurring tannins are considered allelopathic agents. They are constantly released by plants to the surrounding environment and can inhibit the growth and development of nearby plants and other organisms [4].

Flavonoids consist of two aromatic rings connected with a three-carbon bridge; due to this makeup, they are involved in two metabolic pathways: malonate and shikimate pathways. The first step of their biosynthesis is chalcone synthase (CHS) mediated canalization and condensation of a single unit of an ester of hydroxycinnamic acid- CoA (most often p-coumario-CoA) with three units of malonyl-CoA. The resulting 15-carbon chalcone contains two six-carbon rings and most often undergoes isomerization into a flavone by chalcone isomerase (CHI) [29].

Flavonoids are very functionally diverse and can be classified into three groups: anthocyanins, flavonols, and flavones. Most anthocyanins occur as pigments in angiosperm plants whose main function is to attract pollinator insects. They also provide a level of protection against UV light. They are stored in vacuoles, and their color depends on the concentration of $H^+$ ions. Generally, they are red or purple in low pH ranges but turn blue in high pH. Similar to other metabolites described above, anthocyanins are biologically active with potential medicinal use, mostly due to their antioxidant and antimicrobial activities. They may also prevent cardiovascular disorders and complications in the treatment of diabetes. Flavones and flavonols are also pigments and may modify the color of anthocyanins [4]. Their colors range from cream to yellowish, but they also absorb in the ultraviolet spectrum, which, similarly to anthocyanins, gives them ability to protect plants and trees from harmful UV light [4,30].

The last of the large groups of secondary metabolites are non-protein nitrogen compounds: alkaloids, betalains, cyanogenic glycosides, and glucosinolates. These compounds originate mostly from amino acids and their derivatives, but they also include glycosides containing sugar groups [31]. This class of secondary metabolites also includes some non-proteinogenic amino acids, such as L-arginine; analog canavanine can be harmful for animals [32].

Initially, it was suspected that the main role of alkaloids in plants is nitrogen storage. More systematic research, however, showed that due to their high toxicity they are very important in terms of protection against pathogens and insect pests. These biologically

active properties make many alkaloids particularly useful for medicine where they are used in small doses to treat numerous conditions [33].

Betalains are alkaloid plant pigments whose characteristic structural feature is the occurrence of a sugar molecule linked via a glycosidic bond. The betalain-producing plants do not produce anthocyanins, though they may contain other flavonoids. They function as yellow, red, orange, and purple pigments, mostly in flowers but also in other organs. Their role likely also includes protection against pathogens. Cyanogenic glycosides, on the other hand, are typical protective substances. Though harmless in their intact form, they are a rich source of toxic hydrogen cyanide that may be released under stress, thus providing a potent protective mechanism. Hydrogen cyanide is a strong inhibitor of cytochrome oxidase, a key enzyme in respiratory electron transport, and its occurrence disrupts cellular respiration. Enzymatically controlled NCH release occurs only in reaction to injury. Amygdalin is a popular cyanogenic glycoside occurring in tissues of almond trees (*Prunus amygdalus*), which gives it the very characteristic smell of bitter almonds [34]. Interestingly, in many cyanogenic plant species a characteristic distinction occurs between cyanogenic and non-cyanogenic phenotypes [4].

Glucosinolates (also called mustard oil glycosides) are built of a glucose molecule and an aliphatic or an aromatic radicle, connected via S-glycosidic bonding. Similar to cyanogenic glycosides, glucosinolates do not occur in the same cells as the enzymes necessary to hydrolyze them. This ensures the release of a deterrent agent, an isothiocyanate group, occurs only upon an injury of tissue [35].

### 3. Secondary Metabolite Extraction Methods from Plant or Fungal Tissues

Trees are sometimes metaphorically called biochemical factories for the production of primary and secondary metabolites [36]. However, a variety of biotechnological methods are used for the industrial production of secondary metabolites (Figure 3).

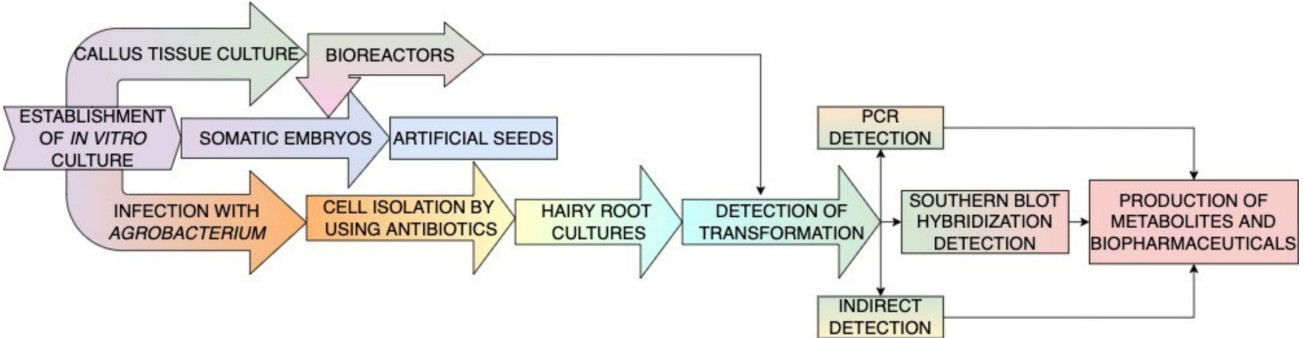

**Figure 3.** Production of secondary metabolites in plant cultures in vitro.

The metabolome analysis, that is, identification of all secondary metabolites produced by a species in genetically diverse populations, vastly facilitates the identification and tracking of metabolic phenotypes [5,12].

The initial step of the production of secondary metabolites in vitro is careful selection of tree material, which is required to ensure high productivity of cultures. The procedure may also require additional differentiation of tissues in cultures due to the sometimes-localized production of the desired compound. In such cases, not all tissues or organs (e.g., hydathodes and oil cells) are specialized in biosynthesis and/or storage of secondary metabolites such as ethereal oils [37]. Moreover, the production of target compounds can be biochemically or physiologically stimulated under strictly controlled conditions in vitro, which is a very useful tool for studying metabolic pathways. This very level of control over the growth conditions can be developed in tree tissue cultures to ensure the production of exactly the same set of metabolites as in nature [38].

An important step in preparing tissue cultures for metabolite production is extraction of these chemical compounds.

A team from the Federal University of São Carlos (2019) developed three key techniques for the production of herb and conventional drugs in vitro that differ in processing of the initial plant material. The first technique relies on the multiphase propagation of plant material in vitro, especially the production of large numbers of highly productive seedlings. The main disadvantage of this approach is its low time-effectiveness; a great deal of time is necessary from initiation of cultures to effective production of target metabolites. The second method relies on the application of a variety of biotic and abiotic treatments, that is, elicitors, in order to stimulate metabolite production in the initial biomass. The last of the methods mentioned above involves genetic modification of the plant species to serve as bioreactors for the production of the desired secondary metabolites. This approach aims to improve the levels of metabolite production in specific plant tissues. The advantage of the latter two methods over the first one is their fast applicability [39]. Another frequently used in vitro method useful in production of secondary metabolites are plant cell cultures. This approach benefits mostly from its simplicity, reliability, and predictability of the underlying processes [40]. Another method used to produce secondary metabolites is hairy root cultures. This method relies on genetic transformation using RI plasmid (root-inducing plasmid), naturally occurring in *Agrobacterium rhizogenes* bacteria [41]. Despite its undoubtful advantages, the method is relatively inefficient, as well as biosynthetically and genetically unstable in the long term cultures [37,42–45].

Introduction of DNA recombination methods on the market enabled the obtaining of secondary metabolites on a larger scale compared to traditional in vitro cultures [46,47]. The quality and yields of secondary metabolite extracts obtained with in vitro methods depends on many factors, especially the quality of reagents and proper optimization of culturing parameters [48].

*Examples of Research Protocols Used to Detect and Quantify the Secondary Metabolites Using Key Analytical Methods*

The underlying concept of all extraction methods used to separate biologically active compounds is separation of substance, and most of the final secondary metabolite extracts are mixtures of phytochemicals. Among the common separation techniques used in the extraction protocols are thin layer chromatography (TLC), column chromatography (preferably high-performance liquid chromatography (HPLC)) and FLASH chromatography. They are sometimes augmented with nonchromatographic detection methods, e.g., serological tests, phytochemical screening tests, and Fourier transform infrared spectroscopy (FTIR), which enable the accurate identification of compounds along the extraction process (Figure 4) [49].

The TLC method, on the other hand, was used to extract SMs from *Salisapilia tartarea*. Ethyl acetate extracts were prepared from both filtrated mycelium and post-culturing medium, and screened for the presence of secondary metabolites using the TLC technique. The amount of total phenols was determined calorimetrically using Folin-Ciocalteu assay, while to determine total flavonoid content the aluminum chloride colorimetric method was used [50]. The second method, HPLC, is based on the extraction of secondary metabolites from, for example, *Pestalotiopsis fici* mycelium, a fungus isolated from branches of tree plants. *Camellia sinensi* has been conducted according to a different protocol. The fungus was initially grown on *C. sinensi* explants in vitro, on where its spores were collected. The resulting organic phase was centrifuged, and the isolated supernatant was analyzed by HPLC and detected by variable wavelength UV, monitoring at 210, 230, 254, and 280 nm [51]. An example of the HPLC method being used for the identification of secondary metabolites is an experiment of Nawrot-Chorabik et al. (2021) where the authors successfully separated a previously unknown secondary metabolite from a fungus growing in dual culture with ash endophyte *Thielavia basicola* [52]. The next example involves the analysis secondary compounds in *Amphoricarpos autariatus* leaves using FLASH chromatography. The resulting supernatants were analyzed using NMR. Such a technique also made it possible to test the samples for cytotoxicity on human cell lines [53].

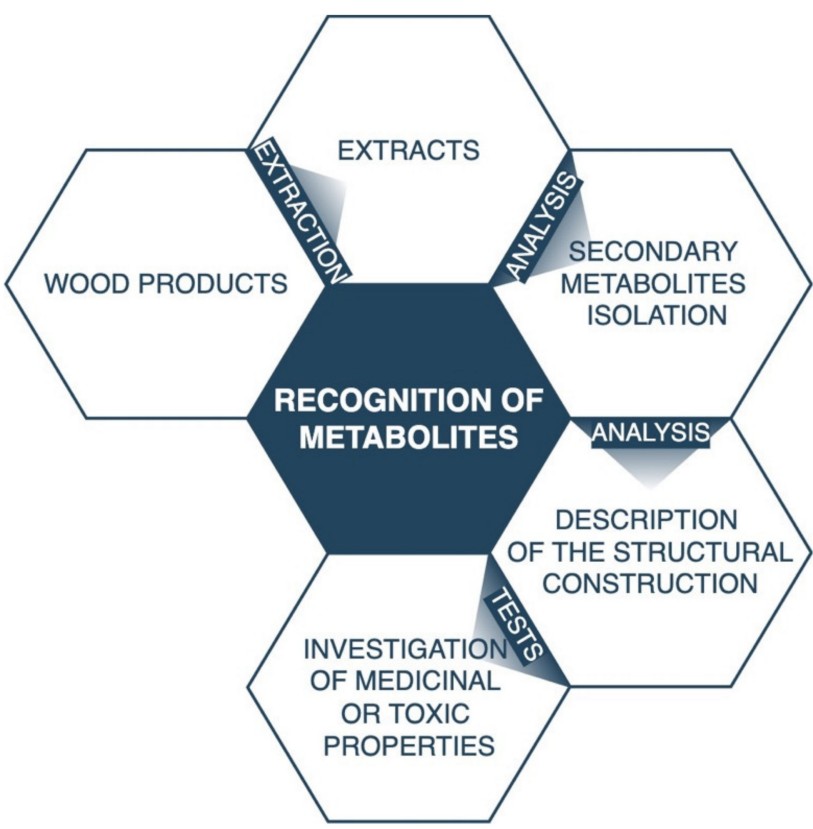

**Figure 4.** A simplified process of identifying secondary metabolites obtained in vitro from wood products.

In recent years, research has been carried out to identify the major phytochemicals in medicinal plants worldwide using methods such as gas chromatography-mass spectrometry (GC-MS) and HPLC. Despite the data obtained on the components of the extracts and essential oils of medicinal plants, due to the lack of in-depth genomic and transcriptomic studies, the biosynthetic pathways and enzymes involved in these pathways remain unknown in most plant species. Therefore, the use of modern Next-Generation Sequencing (NGS) methods has become an accepted approach to the genotype and assessment of the diversity and structure of the population of medicinal plants. NGS methods and their derivative markers give high throughput and reproducible data compared to conventional molecular markers (e.g., RAPD). Precise studies using NGS will not only facilitate the identification of key genes involved in the biosynthesis of compounds, but also provide information for the development of high-throughput molecular markers. Although reference genome sequences are available for most model and popular plants, these data are missing for most medicinal plants. The use of these methods in whole genome sequencing has led to the identification of high-throughput markers, demonstrating their effectiveness and affordability [54]. New computational methodologies may be effective in analyzing metabolic pathways. Recent advances in analytical computational pipelines have enabled the efficient and high-quality exploration and exploitation of single-omics data. However, the integration of multidimensional, heterogeneous, and large data sets remains a challenge. One possible method is Machine Learning (ML). It offers promising approaches to integrating large datasets and recognizing fine-grained patterns and relationships [55].

Genetics is a science widely used in agriculture and pharmacy. It allows for the crossing and modification of plants and trees, through which it is possible to obtain specimens with specific characteristics. Genetic methods include genetic engineering, synthetic biology, and DNA sequencing, among others. Classical genetic engineering involves the use of specific tools, such as PCR or restriction enzymes. Another approach is found in synthetic biology, which uses sequencing and genetic recombination in addition to PCR. Gene synthesis is being introduced in this relatively new field. New components are created from the

already-known genes. Research that can be applied in practice is an incredibly vast field to improve many previously limited scientific paths. Genome sequencing is an equally dynamically developing field. As a consequence, the more precisely the sequences of genomes are analyzed, the more precisely it will be possible to see differences in specific places of DNA.

Genetic research on plants and trees, beyond biological possibilities, has a great economic impact. Understanding the entire metabolome and the characteristics of each relationship of specific trees and fungi will greatly improve medicine and genetic engineering. It will allow humans to develop resistance to pests, diseases, and other stress factors. Due to in vitro breeding, the number of specimens that will be ready for use, e.g., for afforestation, increases in a short time. In the case of the edible parts of trees, they will certainly increase in nutritional value.

## 4. Applications of Secondary Metabolites Occurring in Trees and Fungi

Metabolic pathways in trees are an extensive research area. These long-lived plants produce numerous secondary compounds that are usually necessary for them to function properly. This includes protective compounds whose production developed as an effect of long co-occurrence with numerus pests, pathogens, and other stress factors. There are two types of secondary metabolite-based defense mechanisms: static and induced. The first type involves permanent barriers that are developed in advance, in anticipation of future invasion. The latter type of defense response is activated only after an attack or unfavorable environmental conditions have been detected [56].

Tree populations face new and rapidly changing selection pressures [57]. However, none of the conifer and deciduous tree species have a complete catalog of secondary metabolites. One of the reasons for such a situation may be the great variety of environmental conditions in which trees of the same species occur. Depending on the microclimate and the local populations of organisms associated with trees, especially pathogens and pests, each individual set produces a different set of metabolites. The accelerated production of defensive or protective compounds usually does not begin until the pathological state occurs. Thus, the metabolic response in plants precedes the occurrence of visible disease symptoms [58].

The most readily recognizable secondary metabolites of trees worldwide are α-pinene, β-pinene, and cedrol. The former two compounds are isomers belonging to the monoterpene class. So far, their occurrence has been reported in pines, spruces, firs, and junipers. Due to their broad spectrum of biological activity, both isomers are commonly used in pharmacology and forest therapy. Their medicinal applications can be antiallergic, anxiolytic, anticoagulant, anticancer, and analgesic, among many others. Due to their high stability, they are considered relatively safe compounds for humans [59,60]. Fungi are a rich source of antibacterial, antifungal, antiviral, and antiprotozoal compounds [61–63]. Among the most prominent fungal secondary metabolites are gignardic acid, colletotric acid, viridicatol, and jesterone [64]. Endophytic fungi may produce secondary metabolites with inhibitory effects on plant pathogens [65]. Mycophenolic acid isolated from *Penicillium* sap is a strong immunosuppressive drug preventing organ rejection after transplantation [66].

Secondary metabolites, including those produced in vitro in plant dual cultures, have been studied for their effect on the SARS-CoV-2 virus and their potential use in COVID-19 treatment [67]. An alkaloid, quinine (traditionally used as an antimalarial drug [68]), is an example of such a compound. Chloroquine has been shown to significantly reduce the severity of symptoms caused by coronaviruses due to its biocompatibility [69,70]. A similar coronavirus, SARS-CoV, proved to be susceptible to reserpine, which effectively inhibited its replication. This effect may be used to develop useful antiviral drugs in the future [67,71,72].

The increasing longevity of most human populations results in the increasing significance of age-related conditions including neurodegenerative, cardiovascular, and and neoplastic diseases, which result from multi-factor deterioration of cell functions [73]. In

some cases, the severity of such conditions can be changed by natural compounds with beneficial biological activity. For instance, it has been demonstrated that apple fruits (*Malus* spp. Mill.) have medicinal potential due to the high polyphenol content in apple peel, especially apigenin, which has anti-inflammatory, antispasmodic and antioxidant properties. The compound actively induces apoptosis, and in consequence inhibits the proliferation of breast and ovary cancer cells [73–75]. Similar activity, in addition to their antibiotic properties, has been demonstrated for penicillins and cephalosporins [76,77]. Tetracyclines, that is, a class of antibiotics commonly used in culturing of fungi to prevent bacterial growth, may also be useful in the treatment of prion diseases. These include scrapie disease in sheep and goats, bovine spongiform encephalopathy, Creutzfeldt-Jakob disease, fatal familial insomnia, and Gerstmann-Sträussler-Scheinker syndrome [78–81]. All these potentially transmissible disease cause irreversible neurological degeneration in animals or humans. The anti-prion activity of nontoxic tetracyclines results from their ability to interact with α-helix-rich prion proteins which reduces the rate of their conversion into a β-structure-rich variant. Thus, they may prove the key tools to combat prion diseases in the future [77,78]. Aminoglycosides and macrolides are used to treat various lung diseases and leishmaniosis [77]. Polyene antibiotics actively kill hepatoblastoma cells and inhibit the replication of the HIV virus [77,82], while amphotericin is the only known substance able to inhibit replication of the Japanese encephalitis virus (JEV). Statins, as one of the few secondary metabolites, may support multiple sclerosis [83,84]. According to Cisse and Mucke (2009), paclitaxel (sold under the brand name Taxol), a compound initially harvested from Pacific yew (*Taxus brevifolia* Nutt.), shows an effective destructive activity against breast cancer cells [77,85]. In 1993, the compound was found to occur naturally in yew endophyte *Taxomyces andreanae*, and later in a variety of other yew endophytes. Since then, its antifungal and anticancer properties have been demonstrated [77]. Lastly, cyclosporine A, a commonly used immunosuppressive drug, shows the ability to inhibit intra-erythrocyte growth of the malaria pathogen *Plasmodium falciparum* [86,87].

*4.1. Forest Trees as a Source of Various Secondary Metabolites*

　　Plant secondary metabolites are characterized by high chemical diversity, specific structure, and volatility. Compounds produced by specific metabolic pathways are constantly evolving, adapting the plant to the current internal and external factors, including stress factors [88,89]. With the change of one factor, there is often a change in an element of the metabolic pathway in order to adapt the body to the new conditions as quickly as possible. In forest ecosystems with a high number of plant individuals, fluctuations related to microclimate changes affect their defense, growth, and physiology [90]. Basic climate parameters, such as temperature and humidity, also have a direct impact on insects [91] and pathogens [92], which are two of the main biotic factors influencing the physiology and metabolome composition of host plants [93,94]. They induce the biosynthesis of various secondary metabolites, including phenols, terpenoids, or compounds containing sulfur and nitrogen [95].

　　The occurrence of plant diseases depends on three main stimuli: a favorable environment, a susceptible host, and the presence of the correct pathogen (often with an insect vector) [92,96]. A host that can defend itself against unfavorable conditions will be more resistant to local pathogen genotypes, thus reducing the intensity of the disease (Figure 5) [92].

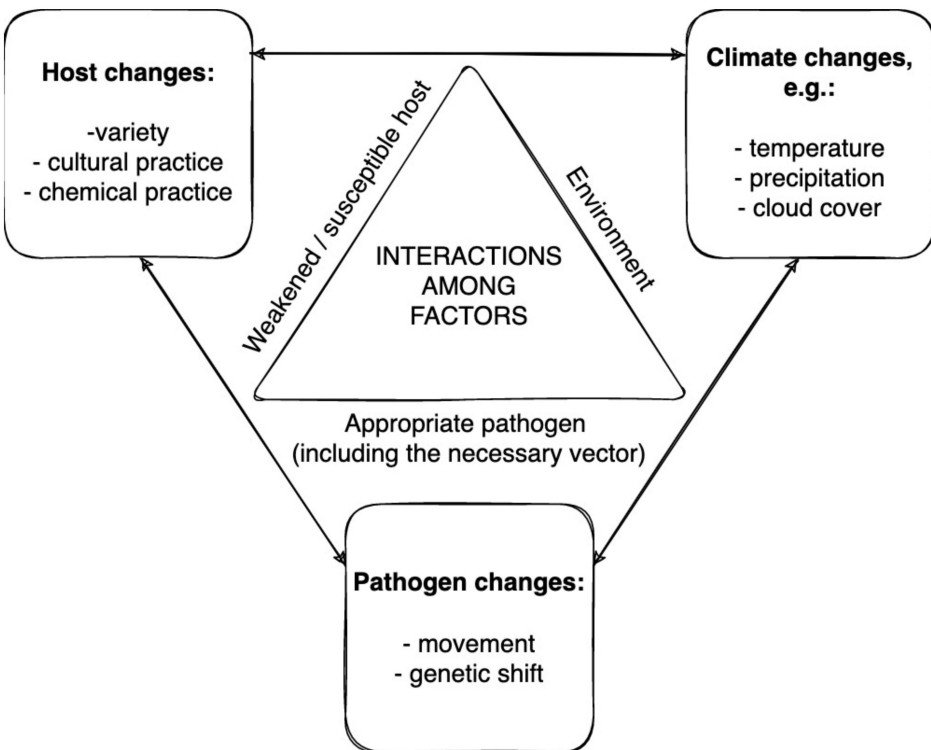

**Figure 5.** Climatic features such as temperature and humidity are important factors in disease. Plant disease consists of three interdependent factors: the host, the pathogen (often with a vector), and the environment. The number of diseases is closely related to changes in these areas.

Like pathogens, there are also fungi that do not take advantage of the host's weaknesses. Most plants create mutualistic symbiosis between their own roots and various mycorrhizal fungi, including arbuscular and ectomycorrhizal fungi [97]. Through close contact between symbiotic partners, fungi obtain carbon, which is a product of photosynthesis, and in return they multiply the plant's access to water and transfer minerals (including nitrogen and phosphorus). According to reports, such a system may also increase tolerance to abiotic and biotic stress, e.g., pests and pathogens [98–102]. Moreover, the mutual symbiotic signals between the partners were characterized [103]. The roots of the host plant actively secreted strigolactones, which are detected by arbuscular fungi and stimulate their spore germination or respiratory metabolism [104]. Many studies have shown that arbuscular mycorrhizal fungi can modify the metabolism of terpenoids and the shikimic pathway, thereby enhancing the biosynthesis of isoprenoids, polyketides and polyphenols. Symbiosis may also increase the content of, inter alia, polyphenols, carotenoids, and the activity of antioxidant enzymes [105–107]. The studies conducted so far suggest a positive role of mycorrhizal symbiosis in the production of beneficial phytochemicals by plants [107].

Interactions between organisms shape the microclimate of the habitat, thus increasing the resources available to other consumers. The above examples of plant adaptation strategies show how the pool of secondary metabolites in a specific ecosystem can grow. Gymnosperm and angiosperm tree species have a largely unexplored pool of metabolites. They are necessary to defend against bacterial, viral and fungal pathogens [108], and to deter herbivores [109,110]. Due to their properties in the natural environment, plant compounds can be successfully used in medicine. Most of the compounds from a given species have their own unique features and purpose [77].

The same species of organisms living in different ecosystems may differ in the composition of the metabolome. Interactions between microorganisms and host trees can therefore create unique microbial communities depending on location [111]. A potential area of research is devoted to further understanding how the interactions between the tree and the microbiome occur and their importance. As a consequence of these interactions, the pool of

secondary metabolites in ecosystems continues to increase, while expanding the metabolic capacity of host plants, bacteria, and fungi. Knowledge of these relationships would not be possible were it not for constantly evolving technologies [112]. Therefore, it is crucial to learn about the relationships between organisms. One of the newer technologies exploring this aspect is multi-omics studies that determine the relationships between host and microbiome elements [113]. Pearson's and Spearman's correlations providing information on factors of metabolic change are also increasingly used to analyze relationships, where operational pairs of taxonomic units and metabolite profiles are used [112,114]. Increasingly, however, studies are no longer using a single method, but conclusions are based on the use of several tools in a single analysis. This approach allows the integration of multiple pieces of information seen from other perspectives [115]. Thus, it seems necessary to strive for the integration of computational and biological approaches, which will allow for increasingly rapid development not only of research, but also of the science of interactions between organisms and secondary metabolites [116,117].

### 4.2. Secondary Metabolites Naturally Occurring in Conifers

A variety of secondary metabolites has been identified in gymnosperm species to date, of which only a small fraction has been characterized biochemically and functionally (Table 1).

**Table 1.** Secondary metabolites produced by gymnosperm species.

| Tree Species | Isolated From | Structural Classification | Examples of Secondary Metabolites | References |
|---|---|---|---|---|
| *Pinus sylvestris* | Needles, wood | Monoterpenes | α-Thujene, α-Pinene, Camphene, β-Pinene, Limonene, Terpinolene | [57] |
| | | Sesquiterpenoids | β-Caryophyllene, β-Copaene, α-Cadinol, Aromadendrene | |
| | Bark, cones | Flavonoids | Catechin, Epicatechin, Epigallocatechin gallate, Verbascoside Caffeic acid | [118,119] |
| | Buds | Non-protein amino acids | Cystathionine, β-Alanine, β-Aminobutyric, γ-Aminobutyric, Ornithine | [120] |
| | Needles | Proanthocyanidins | Prodelphinidins and Propelargonidins catechin derivatives | [121] |
| | | Phenolic acids | Caffeic acid, Salicylic acid, Ferulic acid, Vanillic acid, Gallic acid, Sinapic | |
| *Pinus mugo* | Needles | Monoterpenes | α-Pinen, β-Pinen, Mircen, α-felandren, Terpinolene, Linalol | [122] |
| | | Diterpenes | α-Cadinol, abietatriene, Dehydroabietol, Pimaradiene | |
| *Pinus pumila* | Cones | Flavonoids | Chrysin, Luteolin, Quercetin, Taxifolin, Dihydromyricetin | [123] |
| *Pinus banksiana* | Wood with knots | Flavonoids | Pinobanksin, Pinocembrin, Taxifolin, Naringenin, Dihydrokaempferol | [124] |
| *Pinus nigra* | Foliage | Monoterpenes | β-phellandrene, α-Pinene, β-Pinene, Camphene, Myrcene, Limonene, Terpinene, Linalool | [125] |

**Table 1.** *Cont.*

| Tree Species | Isolated From | Structural Classification | Examples of Secondary Metabolites | References |
|---|---|---|---|---|
| *Pinus yunnanensis* | Foliage | Flavonols | Taxifolin derivative | [126] |
| | | Lignans | Erythro-1-(4-hydroxy-3-methooxy-phenyl)-2-[2-hydroxy-4-(3-hydroxypropyl) phenoxy]-1,3-propanediol, threo-1-(4-hydroxy-3-methooxy-phenyl)-2-[2-hydroxy-4-(3-hydroxypropyl)henoxy]-1,3-propanediol | |
| | | Diterpenes | Iso-Cupressic acid, Agathic acid, Pinifolic acid, Agatholic acid, Agathic acid, 15-methyl ester | |
| *Pinus roxburghii* | Bark | Flavonoids | Taxifolin, Taxifolin derivative, Quercetin, Catechin | [127] |
| | | Lignans | Pinoresinol, Secoisolariresinol | |
| | | Phenolic acids | Protocatechuic acid | |
| | | Stilbenes | Monomethyl Pinosylvin | |
| *Pinus wallichiana* | Bark | Flavonoids | Quercetin, Taxifolin derivative, Catechin, Catechin, Gallocatechin derivative | |
| | | Stilbenes | Monomethyl Pinosylvin, Dihydro-monomethyl Pinosylvin | |
| | | Lignans | Secoisolariresinol | |
| | | Phenolic acids | Protocatechuic acid | |
| *Pinus gerardiana* | Bark | Flavonoids | Taxifolin, Taxifolin derivative, Quercetin, Catechin | |
| | | Phenolic acids | Protocatechuic acid | |
| | | Stilbenes | Dihydro-monomethyl Pinosylvin | |
| *Pinus kesia* | Bark | Phenolic acids | Caffeic acid, Gallic acid, Chlorogenic acid | |
| *Pinus merkusii* | Bark | Stilbenes | Pinosylvin monomethyl ether, Pinosylvin dimethyl ether | |
| | | Flavonoids | Pinocembrim | |
| *Picea abies* | Needles | Phenolic acids | Shikimic acid, Galusic acid, p-Coumaric acid, Protocatechuic acid, Ferulic, Vanillic, Syringic, Sinapic, Salicylic, Quinic acids, Protocatechuic, Gallic acids | [121] |
| | | Flavonoids | Catechin, Kaempferol 3-glucoside, Naringenin, Quercetin, Quercetin 3-glucoside, Quercitrin Catechin, | [128] |
| | | Stilbenes | Cis-astringin, Trans-astringin, Trans-piceatannol, Cis-piceid, Trans-piceid, Trans-resveratrol | [121] |

**Table 1.** *Cont.*

| Tree Species | Isolated From | Structural Classification | Examples of Secondary Metabolites | References |
|---|---|---|---|---|
| | Branches | Monoterpenes | α-Pinen, β-Pinen, Limonen, Myrcene, Limonene, γ-Terpinene, Geraniol | [129] |
| | Emission, needles, xylem, bark, wood, roots | | Linalool, Camphor, Borneol, Piperitone, β-pinene, Terpinolene, α-pinene, Camphene, p-Cymene | [130] |
| | | Sesquiterpenes | β-Caryophyllene, Longifolene | |
| *Picea jezoensis* | Needles, bark, wood | Stilbenes | Trans-astringin, Cis-astringin, Trans-piceid, Trans-piceatannol, Trans-resveratol, Cis-isorhapontigenin | [131] |
| *Abies alba* | Branches | Monoterpenes | α-Pinen, β-Pinen, Limonen, Myrcene, Limonene, γ-Terpinene, Geraniol | [129] |
| | Needles | Sesquiterpenes | β-Caryophyllene, α-Humulene, Santene | [132] |
| *Taxus baccata* | Needles, branches | Alkaloids | 10-Deacetylbaccatin III, Baccatin III, Cephalomannine, Taxinine M, Taxol A | [133] |

The leaves and bark of Japanese red pine (*Pinus densiflora*) contain a variety of secondary metabolites of potential medicinal significance. In their recent study, Ha et al. (2020) found a total of 26 secondary compounds belonging to glycosides, diterpenes, and flavonoids. Some of them show the ability to suppress cytopathogenic effects caused by viral infections. Flavonoids show anti-influenza properties; the effect is related to the blocking of neuraminidase, one of the viral proteins. The antiviral and anti-inflammatory effect of diterpenes is likely related to their ability to modify the expression of genes involved in the replication of viruses. Thus, the Japanese red pine may be regarded as a source of medicinal substance in the case of influenza epidemics [134]. In addition to antiinflammatory and antiviral properties, pine needle extracts, from *P. morrisonicola* and *P. pinaster*, proved to have antioxidant activity [135]. Similarly, high levels of phenols and flavonoids, as well as a high antioxidant index, can be found in fermented pine needles in ethyl acetate extract [136]. Studies on the antibacterial activity of Norway spruce (*Picea abies*) extracts showed very weak inhibitory effects against Gram-positive bacteria *Staphylococcus aureus*. The earlier analyses [137,138] showed a different trend, which was probably caused by different source material being used for the extraction, that is, sapwood, heartwood, and resin [128]. Particular diterpene acids originating from various conifers have antimicrobial and antiulcer activity [139], as well as cardiovascular effects. An interesting inhibitory effect on the soybean 5-lipoxygenase has been identified for abietic acid naturally occurring in, e.g., larch resin [140,141].

The high flavonoid content in European larch (*Larix decidua*) wood indicates its possible antiinflammatory activity. This effect was studied on a group of sows fed with larch sawdust during pregnancy. The results revealed that the treatment reduced the postpartum rectal body temperature of the test pigs, likely due to the flavonoids contained in the wood. Thus, the supplementation of larch sawdust reduced the risk of diseases, and such a treatment may be a useful way to improve the condition of mammary glands and milk production in pigs [142].

The common juniper (*Juniperus communis*) is a shrub or a small tree whose tissues contain large amounts of aroma oils, invert sugars, natural resins, organic acids, terpene acids, leucoanthocyanidins, alkaloids, flavonoids, tannins, natural gums, lignins, waxes,

etc. This rich combination of secondary metabolites is the reason why it has proven diuretic and anti-diabetic properties. Diet supplementation with juniper essential oil supports the treatment of gastrointestinal and autoimmune disorders [143,144]. Juniper berries also contain α-pinene, which is considered an anti-inflammatory agent [145–147] and reduces the production of pancreatic tumor necrosis factor (TNF-α), as well as some of the interleukins-1β [148]. It has also antiproliferative activity against some types of cancer cells, which indicates its anticancer potential [149–152].

Taxins are a group of diterpene alkaloids naturally occurring in yews (*Taxus* spp.). Taxin B occurs throughout the entire plant and determines the toxicity of yews. Paclitaxel (taxol A), due to its cytotoxic and antitumor activities, is used in the treatment of breast, ovarian, and lung cancers. It is also an essential drug in the second-line treatment of AIDS [153,154]. Another antitumor metabolite occurring in the genus *Taxus* is 10-deacetylbaccatin III, a non-alkaloid terpene that promotes the death of cancer cells [133,155].

### 4.3. Secondary Metabolites Naturally Occurring in Angiosperm Trees

The abundance of secondary metabolites in angiosperm trees seems to be as great as in gymnosperm species (Table 2). For various reasons, including the lengthiness of the research process and the complex structure of some compounds, the proprieties of many metabolites have not been yet characterized. However, many of the compounds whose function has been properly studied are widely used as pharmacological substances.

**Table 2.** Secondary metabolites produced by angiosperm tree species.

| Tree Species | Isolated from | Structural Classification | Examples of Secondary Metabolites | References |
|---|---|---|---|---|
| *Betula pubescens* | Buds | Flavonoids | Kaempferol, Apigenin, Quercetin | [156] |
| *Betula pendula* | | | Kaempferol, Apigenin, Quercetin | |
| *Fraxinus excelsior* | Pollen grains | Monoterpenes | α-Pinen, Sambiene, α-Terpinene, β-Pinene, Linalool, α-Terpineol | [157] |
| | | Sesquiterpene | Calarene, α-copaenem β-cubebene, α-muurolene, T-cadinol | |
| *Fraxinus pennsylvanica* | Phloem | Flavones | Apigenin | [158] |
| | | Glycosides | Ligustroside, Oleuropein, Verbascoside | |
| | | Lignans | Syringaresinol | |
| *Fraxinus mandshurica* | | Glycosides | Ligustroside, Oleuropein, Verbascoside, Calceolarioside A, Esculin, Calceolarioside B, Fraxin | |
| *Fraxinus americana* | | Flavones | Apigenin | |
| | | Glycosides | Ligustroside, Oleuropein, Verbascoside | |
| | | Lignans | Syringaresinol | |

**Table 2.** *Cont.*

| Tree Species | Isolated from | Structural Classification | Examples of Secondary Metabolites | References |
|---|---|---|---|---|
| *Populus tremula* | Bark | Glycosides | Salicis, Salicortin, Salireposide, Gradidentanin | [159] |
| | Buds, foliage | Flavonoids | Kaempferol, Apigenin-4-Me, Chrysin, Galagnin, Pinocembrin | |
| | Foliage | Glycosides | Salicortin | |
| *Populus tremuloides* | Bark | Glycosides | Salicortin, Salireposide, Gradidentanin | |
| | Buds, foliage | Flavonoids | Quercetin, Chrysin, Pinocembrin | |
| | Foliage | Glycosides | Salicortin | |
| *Populus alba* | Bark | Glycosides | Salicortin, Salireposide, Gradidentanin | |
| | Foliage | | Salicortin, Gradidentanin | |
| *Populus nigra* | Bark | Glycosides | Salicortin | |
| | Foliage | | Salicortin | |
| *Populus trichocarpa* | Bark | Glycosides | Salicortin, Salireposide, Vimalin | |
| | Foliage | | Salicortin, Salireposide, Vimalin | |
| *Populus candicans* | Bark | Glycosides | Salicortin, Salireposide, Vimalin | |
| | Buds, foliage | Flavonoids | Quercetin, Luteolin, Myricetin | |
| | Foliage | Glycosides | Salicortin, Salireposide, Vimalin | |
| *Salix alba* | Bark | Glycosides | Salicin, Salicortin, Grandidentanin, Triandrin | |
| *Salix aurita* | Bark | Glycosides | Salicin, Salicortin, Triandrin, Vimalin | |
| *Salix caprea* | Bark | Glycosides | Salicin, Salicortin, Triandrin, Fragilin | |
| *Salix fragilis* | Bark | Glycosides | Salicin, Grandidentanin, Triandrin, Fragilin | |
| | Foliage | | Salicin, Salicortin | |
| *Salix myrsinifolia* | Bark | Glycosides | Salicin, Salicortin, Picein, Triandrin, | |
| | Foliage | | Salicin, Salicortin | |
| *Salix pentandra* | Bark | Glycosides | Salicin, Salicortin, Grandidentanin, Triandrin, | |
| | Foliage | | Salicin, Salicortin | |
| *Salix purpurea* | Bark | Glycosides | Salicin, Salicortin, Salireposide, Grandidentanin, | |
| | Foliage | | Salicin, Salicortin | |
| *Salix repens* | Bark | Glycosides | Salicin, Salicortin, Salireposide, Grandidentanin, | |
| | Foliage | | Salicin, Salicortin | |
| *Salix triandra* | Bark | Glycosides | Salicin, Salireposide, Tremulacin, Grandidentanin, Triandrin | |
| | Foliage | | Tremulacin | |
| *Salix viminalis* | Foliage | Glycosides | Salicin, Salireposide, Triandrin, Vimalin | |

Examples of these compounds include, for instance, betulinic acid, naturally occurring in Betulaceae family. It can also be produced via chemical synthesis from betulin occurring in trees belonging to the same family [160,161]. Both betulin and betulinic acid are known anti-inflammatory compounds [162]. A variety of other pentacyclic triterpenes occur naturally in birch bark [163]. Pure betulin and its extracts have an antiproliferative effect observed in vitro and mainly anti-inflammatory activity in vivo, which determines its anticancer potential [164].

As demonstrated in the study conducted by Tanase et al. (2019) the microwave-assisted extracts from European beech (*Fagus sylvatica*) had antioxidant, antibacterial, antifungal, and antimutagenic effects and an inhibitor effect to tyrosinase and α- glucosidase. The beech bark contained 18 polyphenols: caftaric acid, gentisic acid, caffeic acid, chlorogenic acid, p-coumaric acid, ferulic acid, sinapic acid, hyperoside, isoquercitrin, rutin, myricetol, fisetin, quercitrin, quercetin, patuletin, luteolin, kaempferol, and apigenin [165]. In some cases, the alcoholic extracts (with flavonoids) of *F. sylvatica* can be used to modulate the production of volatile fatty acids. The aim is to enrich the diet of farm animals to solve their health problems and to have a positive impact on the environment [166].

Bird cherry and black cherry (*Prunus padus* and *P. serotina* respectively) are the richest sources of lipophilic triterpene ursolic, corosolic, and oleanolic acids. They have antiinflammatory, anti-ulcer, antioxidant, hepatoprotective, anticancer, antiatherosclerotic, and antidiabetic properties [167–173].

Studies on the pharmacological potential of English hawthorn (*Crataegus laevigata*) are still in the experimental phase. However, it was demonstrated that flavonoids and phenolic acids contained in its extracts may have antinecrotic potential [174,175]. The use of phenolic acid found naturally in hawthorn [176] proved an effective treatment, reducing necrotic changes in rat colon diseases induced by 2,4,6-trinitrobenzenesulfonic acid [175,177–179]. Chinese hawthorn (*Crataegus pinnatifida*), on the other hand, is recognized as a medicinal species by Chinese traditional medicine. Preparations based on this species are used to improve blood circulation and digestion. The *C. pinnatifida* berries, leaves and flowers were found to contain 150 secondary metabolites belonging to flavonoids, terpenoids, oligomeric proanthocyanidins, and organic acids [180–182]. Fresh hawthorn berries contain relatively high amounts of pectin, exceeding 20%, which are responsible for their antioxidant, hypolipidemic, anti-glycatic, and antibiotic effects [180,183–186]. However, the harvesting of hawthorn fruits is not practical [187]. The flowers, leaves, seeds, and berries of azarole and common hawthorns (*C. monogyna* and *C. azarolus*, respectively) are known in some countries for their antispasmodic, diuretic, and antiatherosclerotic properties [180,188,189].

Ethyl acetate and methanol extracts (retaining gallotannins and ellagitannins or flavonol glycosides and methoxylated flavones, respectively) from the common hornbeam (*Carpinus betulus*) leaves and bark were demonstrated to have inhibitory effects against human tumor cell lines [190]. Felegyi-Tóth et al. (2022), on the other hand, for the first time managed to isolate diarylheptanoids from hornbeam tissues, which show cytotoxic, anti-inflammatory, antimicrobial, and antioxidant effects [191,192]. The ongoing studies indicate that various hornbeam species are a rich source of medicinal compounds.

Studies in herbal medicine have shown that the leaves, fruits, and seeds of ash (*Fraxinus* spp.) trees have antiviral, anti-inflammatory, antioxidant, and cytotoxic effects. The metabolites isolated from trees in this genus to date include coumarins, secoiridoids, phenylethanoids, flavonoids, essential oils, and lignans [157,193–196]. Further studies undoubtedly may yield additional active substance, and the work in this field is currently being conducted in Poland at the University of Agriculture in Cracow in cooperation with Jagiellonian University [52]. One of these such experiments involved testing the interaction between *Thielavia basicola*, *F. excelsior*, and *F. pennsylvanica* callus in dual cultures in vitro. One of the combinations of this experiment showed that not only the fungus produced external metabolites, but the tree callus tissue stimulated by an elicitor (the endophytic fungal copartner in dual culture) also reacted in the characteristic way (Figure 6) [52].

Compounds that were isolated during an in vitro dual culture interaction will likely be identified in the future [52,197].

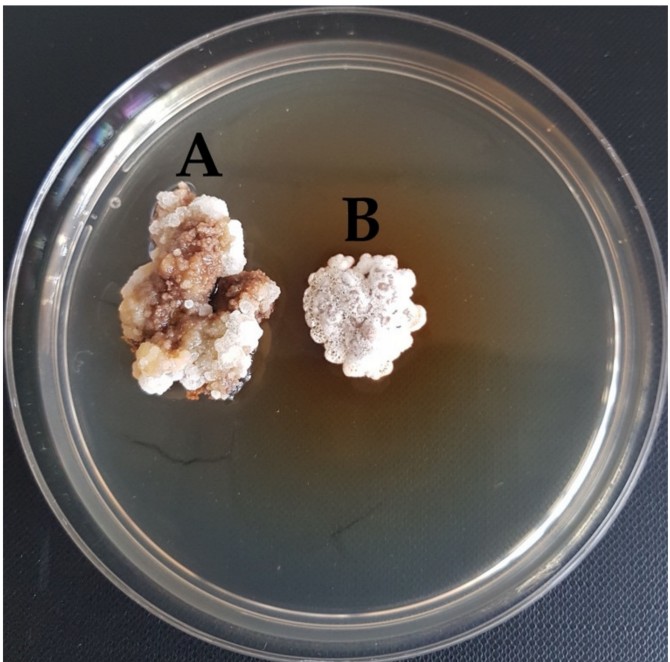

**Figure 6.** Interaction in in vitro dual culture between *Fraxinus excelsior* callus (**A**) and endophyte *Thielavia basicola* (**B**). The fungus did not kill the *F. excelsior* callus after 60 days of culturing without passaging. The mycelium of *T. basicola* developed a distinctive white coating that made the entry of hyphae into the tissues of the callus impossible. It is probable that the growth of mycelium was inhibited by the callus tissue producing secondary metabolites. During culturing the tree tissue remained fully viable and optimally hydrated. The endophyte also caused a yellow staining of the MS medium [52].

Maple syrup, produced from the sap of sugar maple and red maple (*Acer saccharum* and *A. rubrum*, respectively), is a commonly known sugar substitute [198–200]. Beside sucrose, the syrup contains organic acids, monoacids, and various other phythochemicals, mostly phenols [198,200–202]. Li and Seeram (2010, 2011) isolated from maple syrup more than 50 phenolic compounds, including lignans, coumarins, stilbenes, and phenol derivatives [198,203,204]. Phenol-enriched maple syrup extracts proved to have antiinflammatory, antioxidant, antiproliferative, antiradical and antimutagenic activities during in vitro trials [200,205].

Kaempferol-3,7-O-α-dirhamnoside and quercetin-3,7-O-α-dirhamnoside are main flavonoid compounds isolated from the leaves of silver linden (*Tilia argentea*). Both show strong anti-inflammatory and antinociceptive activities in mice without any severe toxicity or stomach damage [206].

Ren at al. (2017) provide detailed information on the secondary metabolites isolated from alders (*Alnus* spp.) [207]. So far, the phytochemical analyses reveal the occurrence of diarylheptanoids, flavones, polyphenols, terpenoids, and steroids, as well as other classes of metabolites, many of which have anti-inflammatory activity. Moreover, the pharmacological potential of diarylheptanoids is extensively studied, due to their tendency to inhibit cancer cells [208–210].

The leaves and bark of the common aspen (*Populus tremula*) contain salicin, salicortin, and tremulacin [211], while black poplar (*P. nigra*) contains mostly phenolic compounds, terpenoids, flavones, and flavanones, as well as ethereal oils [212]. The study of Grigore et al. (2022) confirmed the biologically active character of extracts obtained from poplar buds, which showed antioxidant and anti-inflammatory effects [213].

The Dutch elm disease is (DED) is a wilting disease in elms (*Ulmus* spp.) caused by *Ophiostoma ulmi* and *O. novo-ulmi*. It caused widespread death of millions of elms in Europe, Asia, and in North America [214,215]. The recent studies, however, showed that some secondary metabolites are able to suppress the pathogenic potential of DED pathogens. These included salicylic acid and carvacrol, which supplemented to the water used in irrigation. Elm seedlings showed the strongest effect against *Ophiostoma* spp. in vitro and in vivo. Similarly effective in vitro was thymol, but it proved ineffective in vivo. Application of salicylic acid, monoterpene phenols, carvacrol, and thymol has a significant inhibitory effect on the defensive response of seedlings against *O. novo-ulmi* isolates. This decreased the severity of disease symptoms without elimination of the infection itself [216]. Experiments such as these are examples of biotechnological studies aiming to increase the knowledge on pathogen resistance in elms. They may help to develop effective control measures against DED and to preserve biodiversity of various elm species [217].

### 4.4. Secondary Metabolites Naturally Occurring in Fungi

More than century-long studies on fungal metabolites yielded numerous medicinally important compounds (Table 3), whose widespread uses include treatment of cancer, cardiovascular and neurological disorders, bacterial and fungal infections, and malaria, as well as autoimmunological diseases [218–221].

**Table 3.** The examples of secondary metabolites produced by fungi.

| Species of Fungi | Host Plant | Isolated from | Examples of Secondary Metabolites | References |
|---|---|---|---|---|
| *Cryptosporiopsis* cf. *quercine* | *Tripterygium wilfordii* | Stems | Cryptocin | [222] |
| *Pestalotiopsis fici* | unidentified | Branches | Skyrin, Secalonic acid A, Emodin, Norlichexanthone | [223,224] |
| *Talaromyces pinophilus* | *Arbutus unedo* | Branches | Herquline B, 3-O-methylfunicone | [225] |
| *Chaetomium globosum* | *Ginkgo biloba* | Leaves | Gliotoxin, epipolythiodioxopiperazine | [226,227] |
| *Hormonema* sp. | *Juniperus communis* | Leaves | Enfumafungin | [228] |
| *Sordariomycete* sp. | *Euconia ulmoides* | Leaves, roots | Chlorogenic acid | [229] |
| *Alternaria brassicicola* | *Mallus halliana* | Leaves | Alternariol 9-methyl ether, altechromone A, herbarin A, cerevisterol, 3b,5a-dihydroxy-(22E,24R)-ergosta-7,22-dien-6-one | [230] |
| *Fusarium avenaceum* | *Abies balsamea* | Foliage | Enniatin A | [231] |

The most readily noticeable example of these are *β*-lactam antibiotics, that is, penicillins and cephalosporins, whose significance for medicine and human society is difficult to overestimate [232]. No less significant, but in a negative way, are various mycotoxins produced by fungi. This structurally diverse group of compounds includes many classes: aflatoxins, trichothecenes, fumonisins, ochratoxins, and cytochalasins [233]. This very variability and ubiquitous occurrence in nature makes the studies of mycotoxins a very wide research area. Proper understanding of the toxicity mechanisms and mycotoxin-related interactions between organisms may in the future help to more effectively prevent the threat of mycotoxin contamination and develop useful applications for these compounds. Mycotoxins, as well as other phytotoxic substances produced by fungi, are closely related to many diseases among crops and forest trees, thus they cause great economic loss in numerous sectors of plant production. This means that an effective control of mycotoxic fungi is a very important issue [234].

Only a fraction of secondary metabolites of fungal origin have been functionally characterized. However, these organisms become a valuable source of some bioactive substances. Fungi developed robust metabolic mechanisms enabling them to produce compounds with very variable structural and bioactive characteristics. These substances often offer them advantage in their very variable environmental niches, including some very competitive ones, e.g., soil. This coincidently created a very rich source of biologically active compound useful for medicine [235].

The first discovered secondary metabolite in fungi was mycophenolic acid, identified in 1896 in *Penicillium glaucoma* by Italian physician Bartolomeo Gosio [235]. Identification in 1929 by Alexander Fleming of another antibiotically active compound, penicillin, started the ongoing pursuit of biologically active substances in fungi. This very compound, originally discovered in *P. notatum*, still saves millions of lives every year [236]. Presently, the most important fungi-derived pharmacological products include: antibacterial, antifungal, and antiparasitic drugs, immunosuppressants, and hypolipidemic agents, as well as antibiotics with anticancer activity [68,237].

Another compound isolated from *Penicillium* fungi is a statin mevastatin (also known as compactin) naturally occurring in *Penicillium citrinum* [238]. Statins are currently in clinical use, three of which occur naturally. These, apart from mevastatin, are lovastatin found in *Aspergillus terreus* [239] and pravastatin in *Streptomyces carbophilus* bacteria [240]. Statins are used as hypolipidemic agents reducing the levels of total cholesterol and triglycerides. Such a treatment reduces the risk of cardiovascular disease but may also increase the risk of diabetes [241].

The most widely known immunosuppressive drug is cyclosporin, extracted from *Trichoderma* spp. and *Tolypocladium* spp. fungi. They reduce the activity of T cells by selectively binding to cyclophilin protein. The cyclosporin-cyclophilin complex inhibits the activity of transcription factors regulating the expression of inflammatory cytokines. Cyclosporin is an essential drug preventing post-transplant organ rejection, as well as being used for the suppression of autoimmunological diseases [237,242].

Various types of cancer are some of the main causes of premature deaths in the world. Thus, the search for new cytotoxic and antiproliferative compounds that would help combat the cancer epidemic is of great importance. This also concerns fungal secondary metabolites, examples of which are compounds occurring in *Cordyceps* or *Shitake* fungi. *Cordyceps sinensis* (*Cephalosporium sinensis*), for instance, is a species used in Chinese traditional medicine, with proven anticancer effects. The 2018 studies examined the antimetastatic effect of water extracts of this fungus [243,244]. The metabolites of *C. sinensis* also include a low-activity cytostatic compound and sterol derivatives that inhibit the proliferation of particular human cancer cell lines [245].

An interesting example of a secondary metabolite producer is *Thielavia basicola*, an endophytic fungus occurring in European ash *Fraxinus excelsior* tissues. It showed strong inhibitory effects against the European ash pathogen *Hymenoscyphus fraxineus*. The readily visible sign of this defensive action was brown discoloration of the growing medium and production of dark crystalline structures in the vicinity of the endophyte's colonies (Figure 7) [52]. Compounds involved in these interactions will likely be identified in the near future, and their potential usefulness for medicine or for plant protection will be investigated [52].

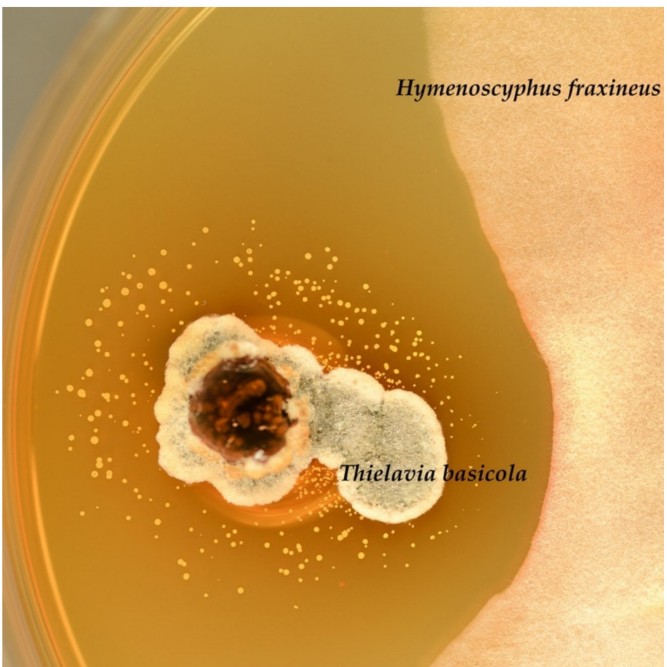

**Figure 7.** Interaction in in vitro dual culture between *Hymenoscyphus fraxineus* (pathogen) and *Thielavia basicola* (endophyte). Macroscopic observations of the interaction between *H. fraxineus* and *T. basicola* show a strong inhibitory effect of the endophyte on the growth of the pathogen. In dual cultures, the endophyte produced an unidentified secondary metabolite secreted into the medium, visible in the form of dark beige crystal-like structures [52].

## 5. Conclusions

Secondary metabolites of various origins undoubtedly have great potential for scientific study. The common use of new analytical and preparative techniques to obtain them will allow for more and more extensive analyses aiming to characterize most of the compounds. In vitro cultures of plant cells, hair roots, or recombinations of DNA are promising methods of their extraction. Among the more promising of these methods are plant tissue cultures in vitro, hairy root cultures, and DNA recombination techniques. The ability to adjust the metabolic pathways via genetic transformation of microorganisms introduces the opportunity to constantly increase the number of available compounds. This, however, requires correct and comprehensive understanding of the biosynthesis process of secondary metabolites. Knowledge in this field will not only help to improve plant production but may also lead to the development of new products containing secondary metabolites. These, of course, may include substances with very useful applications, such as new drugs or new pharmacological processes. Secondary metabolites already provide a selection of very useful drugs essential in treating e.g., diabetes and cardiovascular disorders, as well as bacterial and fungal infections. Numerous reports indicate that well-researched and correctly dosed secondary metabolite-based treatments are not only safe, but also effective. Numerous researchers, as well as medicine practitioners, warn that more and more bacteria can develop resistance to synthetic antibiotics, which becomes a significant problem for modern medicine and for the pharmaceutical industry. The development of novel treatments for bacterial diseases is one of the strategies employed to combat drug resistance. This approach involves searching for new biologically active secondary metabolites, including new antibiotics, in plants and fungi, as well as the adaptation of known compounds for pharmacological use based on their confirmed functions in environmental interactions. Moreover, the development of biotechnological techniques, especially DNA recombination and genetic transformation methods, enables the cost-effective production of organisms rich in secondary metabolites, either plants or fungi, which greatly facilitates the mass production of drugs. Edible vaccines, e.g., tomatoes containing rabies lyssavirus

proteins, are another rapidly developing branch of medicine reducing the material cost and environmental impact of medical treatments. In summary, secondary metabolites obtained from plants and fungi have the potential to become the drugs of the future, thanks to which access to these pharmaceutical products will certainly become more common. Apart from the obvious use of chemical compounds contained in trees, they can be beneficial during forest excursions. Being in a forest environment and breathing the air present there introduces many beneficial substances to the air passages. Multi-species forests contain a very diverse pool of essential oils, so they can be considered the most beneficial for overall human health.

**Author Contributions:** Conceptualization, K.N.-C. and N.G.; validation, K.N-C. and N.G.; writing—original draft preparation, N.G.; visualization, N.G.; supervision, K.N.-C.; funding acquisition, M.S. All authors have read and agreed to the published version of the manuscript.

**Funding:** Article preparation charge was financed by a subvention from the Polish Ministry of Science and Higher Education for the University of Agriculture in Krakow for 2022, No. SUB/040013-D019. Article processing charge was financed by professor Małgorzata Sułkowska.

**Acknowledgments:** The authors thank Tadeusz Kowalski from University of Agriculture in Krakow for providing the photographs of dual fungi culture used in this Manuscript. The authors also thank Dariusz Latowski from the Jagiellonian University in Krakow for scientific cooperation for pioneering research on a synthesized secondary metabolite from material derived from dual in vitro cultures.

**Conflicts of Interest:** The authors declare no conflict of interest.

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
