# Peer review of "Secondary Metabolites Produced by Trees and Fungi: Achievements So Far and Challenges Remaining"

_forests, doi:10.3390/f13081338_

Round 1

Author Response

Dear Professor,

Thank you very much for all your comments on our publication. Each of them was very constructive and resulted in a significant improvement of the Manuscript. The topic we have covered is indeed very broad, as the Reviewer noted, but the main challenge of this work was to concentrate and present this knowledge in a concise form, while still accessible to any potential audience.

The changes proposed by the Professor definitely helped to achieve the intended form of the Manuscript.

Changes we have made to the Manuscript include:

  • Comment 1: „Overall, the content seems general at times due to how broad it is.” – Yes, this is a valuable comment on your part regarding the generality of the work. The topic covered is indeed very broad, as you noted in your review, and the main challenge of this work was to concentrate this knowledge in a concise form at the same time to make it accessible to any potential audience. Therefore, in order to make it more concrete, we tried to follow all your comments, which helped us narrow down some general references. As a result, despite its general nature now, as far as possible, it will point to the relevant topics of the topic covered.
  • Comment 2: „The references listed in the manuscript range from publications that are decades old (more than 35 years that could be considered unnecessary for publication)” – The manuscript will definitely be helped by your proposed changes. Most of the publications that did not contribute theory have been removed or replaced by more current publications. The newer studies will definitely reflect the current issues better. Thirty-five literature items have been removed, along with the content attributed to them.
  • Comment 3: „New techniques (sequencing, synthetic biology, etc.); are slightly brushed in the text, but explaining more about how these technologies are changing science and their impact on the pharmacy and agricultural industries would have been interesting to read more about it and the challenges in that area.” – Thank you very much for another valuable comment on the lack of information on sequencing technologies. We have added a short text on how sequencing and other techniques affect the development of medicine or agriculture, and what can be achieved with them. We hope this makes the text more readable and understandable.
  • Comment 4: „some pictures (the author's inclusion from their experiments) seem cropped and not as clear as the others presented in their work (including their legends)” – The changes you proposed will definitely help in the reception of the manuscript. The descriptions to the photos of the experiments have been expanded, and a legend to one of the photos has been added. As for changing the cropped photo, it was lent to us from Professor Kowalski (University of Agriculture in Cracow; Poland), so changing it could be problematic. If there is a need, the photo will be removed. Thank you very much for your comments, as adding more extensive descriptions to the photos will certainly help to make them more understandable.
  • Comment 5: „but I think that from the microbiome perspective (which They mention briefly in the manuscript) would have been interesting due to the interest in this área and how it links to the new technologies that are being constantly developed.” – Thank you for another valuable comment on the manuscript. In the text, we referred to the microbiome and its interactions with trees. At the same time, we pointed out what technologies to study these interactions are used today. This allowed us to expand the topic and reach new publications from the topic.
  • Comment 6: „Due to how broad the topic is, a section such as challenges (a part of the title of the manuscript) is not as prominent in the text because all the details of all the other different topics are occurring.” – The next valuable comment from your side was the one about the challenge section. Indeed, we did not include it in the original version of the manuscript, which has been corrected. We highlighted a topic that had not yet been addressed, namely interactions between bacteria and fungi to induce resistance in trees, and included it in the Introduction section.
  • Comment 7: „Legend of figure 2 is not the same as the description in the text (there are only three main types of metabolites, not four)” – Thank you very much for pointing out the error. It has been corrected from "four" to "three".
  • Comment 8: „repetitive word (structural structure)” – Thank you for pointing out the error. We left the structure alone.
  • Comment 9: „Figure 6 is from the authors, but the legend and description could be better. The citation is in the text, but since it uses numbers, it will be helpful to include in the figure legend that it is from their group.” – A better perception of this manuscript was certainly helped by the changes made after this comment. A legend has been added to the photo and the description has been expanded so that the description better shows what is depicted in the photo.
  • Comment 10: „-Table 3 descriptions (title) and presentation could be improved. Their title says -"Secondary metabolites produced by fungi"-, and later in the table say some examples of metabolites. Maybe the table could be the highlight of some of the most used/potential application metabolites (either due to their use or production)” – Thank you for further valuable suggestions. The description of the table has been corrected indicating that these are only examples. As for the examples in the table, they are only examples of secondary metabolites produced by fungi. The most relevant ones, in our opinion, have already been included in the text below the table and their very important use has been added. Thank you very much for your comments, as they will certainly help to make the publication better.
  • Comment 11: „-bun (change to but)” – Thank you very much for finding this error. It has been corrected to "but".

We have corrected all your comments in the manuscript. We have included links to the following points in the review where we have made changes based on the reviewer's suggestions. Once again, we thank you very much for each individual comment and hope that in its current form the paper will be reviewed favorably.

Reviewer 2 Report

I agreed to review this paper because of its title. I was expecting to read about the natural products (i.e. secondary metabolites) that have been isolated from TREES. Secondly, I assumed that I would read a review about endophytes found in TREES. As I suspected, this review was attempting to cover an area of research that is extremely large. I felt it came up short. Further, the review discusses various analytical and extraction techniques that are quite common and really do not need to be discussed. I believe all of the analytical techniques could be eliminated from this review. The paper seems to jump from “plant” to “trees” quite often. Use “tree tissue culture” not plant tissue culture, if the intent is to talk about secondary metabolites from trees. The discussion on fungi should be restricted to endophytes found exclusively (if possible) in trees. The discussion about penicillin seems to be off-target.

This paper is both well written and referenced by the authors. I just find the contents of the paper too broad and thus rarely does it offer a deep review of a particular subject. As a natural products chemist I would find this paper of very little use. I think it could still be published but its focus should be TREE and TREE ENDOPHYTE secondary metabolites. I would also suggest to leave out the various analytical methods.

Author Response

Dear Professor,

Thank you very much for all the comments on our publication. Each of them was valuable and allowed us to clarify many aspects of the Manuscript, making it much improved. Indeed, the publication is intended to cover a large area, as you noted in your review, but the goal was to concentrate and present this knowledge in a concise form, while still accessible to any potential audience.

The changes proposed by the Professor definitely helped to achieve the intended form of the Manuscript.

Changes we have made to the Manuscript include:

  • Comment 1: „I was expecting to read about the natural products (i.e. secondary metabolites) that have been isolated from TREES” and „I think it could still be published but its focus should be TREE and TREE ENDOPHYTE secondary metabolites” – Thank you very much for this comment. However, we have included pathogenic fungi in this manuscript, but in an abbreviated form. We motivate our decision by the fact that pathogens also interact with trees and are producers of secondary metabolites and a very important component of the ecosystem. Therefore, in Table 3, we have included endophytes in the first places, followed by fungi from the other trophic groups. We thank you for your comments and hope that the current form will be equally valuable to the reader.
  • Comment 2: „ As I suspected, this review was attempting to cover an area of research that is extremely large. I felt it came up short.” – We have also taken this valuable suggestion from you to pay attention to the generality of the manuscript. Indeed, due to the broad subject matter, it sounds quite general, but we also wanted to describe it comprehensibly to people who have not yet been exposed to secondary metabolites. Thank you for your feedback.
  • Comment 3: “Further, the review discusses various analytical and extraction techniques that are quite common and really do not need to be discussed” – The commentary on the analytical and extraction techniques found there is, according to us, very specific. Indeed, the methods are well known, which we agree with, so following your suggestion we have left only a brief outline of the techniques. For those who are familiar with the subject matter, i.e. chemists and biochemists, it is of course not necessary to explain what the specific methods consist of. The information we have left abbreviated to a minimum on techniques and analysis will perhaps become interesting to the novice. For each method, we left a maximum of 3 sentences of description without elaborating on the entire method and omitting the reagents. We thank you for this comment, as it helped us to make the work more concrete.
  • Comment 4 thematically linked to comment 3: „I believe all of the analytical techniques could be eliminated from this review.” and „ I would also suggest to leave out the various analytical methods.” – Thank you for these comments on analytical techniques. Here, too, we made the assumption that not everyone who comes to our publication will be familiar with the subject matter. Sentences have been reduced to a maximum of 3 per technique, and some have been removed. So, thanks to these comments, we excluded unnecessary elements of descriptions and shortened the manuscript a bit.
  • Comment 5: „The paper seems to jump from “plant” to “trees” quite often.” – The better reception of the work was definitely helped by this comment from the Professor. After analysis, we decided that mixing plants and trees in the text was incorrect, so the information about compounds from plants was removed to focus more on trees only. Thank you for your constructive comment, which will certainly help to improve the reception of the manuscript.
  • Comment 6: „Use “tree tissue culture” not plant tissue culture, if the intent is to talk about secondary metabolites from trees.” – Thank you for your valuable comment. In accordance with it, we have changed "plant tissue cultures" to "tree tissue cultures" because not only does it sound better, but it is more relevant to the topic of the manuscript.
  • Comment 7: „The discussion on fungi should be restricted to endophytes found exclusively (if possible) in trees.” – Thank you for this comment. In this paper, we focused not only on endophytes, but also on fungi from other biotrophic groups to show their important role in the ecosystem. They are also producers of secondary metabolites, which is why they were included in the review. However, we have ranked the endophytes on the top rows in table number 3 to emphasize the fact that they are not as destructive as pathogens, for example. As suggested, we removed the endophytes found on plants leaving only those found on trees. Thank you very much for your comment.
  • Comment 8: „The discussion about penicillin seems to be off-target.” – With this remark, the reception of the manuscript can certainly improve. The chapter on penicillin was far too long, thus overshadowing other fungi. We have shortened it to its basic form leaving a few key sentences. Thank you very much for bringing it to our attention, because after your suggestion, the penicillin issues were discussed in a more logical way.

We have tried to correct or respond to all your comments on the original version of our manuscript. In the body of the manuscript, we have included links to subsequent review points where we have made changes. Once again, we would like to thank you very much for each individual comment and hope that in its present form the paper will be reviewed favorably.
